# The Effect of Community-Based Exercise on Health Outcomes for Indigenous Peoples with Type 2 Diabetes: A Systematic Review

**DOI:** 10.3390/ijerph21030290

**Published:** 2024-03-01

**Authors:** Lauren Hurst, Morwenna Kirwan, Vita Christie, Cara Cross, Sam Baylis, Liam White, Kylie Gwynne

**Affiliations:** 1Department of Health Sciences, Macquarie University, Talavera Road, Sidney, NSW 2109, Australia; lauren.hurst1@hotmail.com (L.H.); morwenna.kirwan@mq.edu.au (M.K.); sam.baylis@students.mq.edu.au (S.B.); liamtwhite8@gmail.com (L.W.); 2Djurali Centre for Aboriginal and Torres Strait Islander Health Research, Heart Research Institute, Eliza Street, Newtown, NSW 2042, Australia; kylie.gwynne@hri.org.au; 3DVC Indigenous Office, University of New South Wales, High Street, Sidney, NSW 2052, Australia; cara.cross@unsw.edu.au

**Keywords:** Indigenous, type 2 diabetes, exercise, community-based

## Abstract

Indigenous peoples globally experience a high burden of type 2 diabetes in comparison to non-Indigenous peoples. While community-based exercise interventions designed for type 2 diabetes (T2D) management have garnered success in non-Indigenous populations, they likely require adjustments to meet the needs of Indigenous people. This systematic review aims to determine if health outcomes in Indigenous peoples with T2D could be improved by community-based exercise programmes and the features of those programmes that best meet their needs. The CINAHL, Embase, Informit Indigenous Collection, Medline, PubMed, Scopus, SportDiscus, and Web of Science databases have been searched to identify peer-reviewed literature with original outcome data that report on the health effects of community-based exercise interventions for the management of T2D among Indigenous peoples. The Mixed Methods Appraisal Tool and Indigenous Community Engagement Tool were implemented to assess methodological quality. Three moderate-to-high-quality studies were selected for review, including participants of Polynesian or Native American Zuni Indian descent. Results indicated positive effects of group exercise on glycated haemoglobin (HbA1c), body mass index, body weight, total cholesterol, blood pressure, quality of life, and patient activation levels in high-adhering participants. This review concludes that community-based exercise interventions may improve health outcomes for Indigenous adults with T2D when conducted with strong community engagement.

## 1. Introduction

On a global scale, Indigenous peoples experience a high burden of chronic disease and health outcome inequalities [1,2,3]. These inequalities arise from the socioeconomic disadvantage that originated with colonisation and are perpetuated by the ongoing discrimination and marginalisation faced by Indigenous peoples [1,4,5,6]. The impact of colonisation on Indigenous communities is so substantial that it is recognised by the World Health Organisation as the most significant social determinant of health for Indigenous peoples worldwide [6,7]. This includes, but is not limited to, Aboriginal and Torres Strait Islander peoples in Australia; Māori and Pacific peoples in New Zealand and the Pacific Islands; American Indian and Alaskan Native peoples in the United States of America; and First Nations, Inuit, and Métis peoples in Canada [6,8,9,10,11,12]. For these populations, the introduction of the Western diet, tobacco, and a sedentary lifestyle has increased their risk of developing lifestyle-related chronic diseases such as type 2 diabetes (T2D) [7,13,14]. Despite this, Indigenous peoples continue to overcome adversity and maintain connections to community, culture, land, and ancestry [15,16,17].

Diabetes is the fastest-growing chronic disease worldwide, with one in every eleven adults currently having the condition [2,18]. This number is disproportionately higher for Indigenous peoples. For example, in New Zealand, Polynesian peoples are three times more likely than non-Indigenous peoples to have diabetes [19]. In 2019, diabetes was the direct cause of 1.5 million deaths and the indirect cause of millions more due to diabetes-related kidney disease and cardiovascular deaths as a result of elevated blood glucose [18,20]. In the same year, diabetes mortality rates for American Indian/Alaska Native communities in the United States of America (USA) were 3.2 times higher than for other American populations [21]. 

The three main types of diabetes are type 1, type 2 (T2D), and gestational diabetes, of which approximately 90–95% of cases are T2D [2,20,22,23]. T2D significantly contributes to the substantial disease burden among Indigenous populations worldwide. It is noteworthy that more than 50% of Indigenous peoples aged 35 and above are affected by this condition [7,24]. T2D involves the body becoming resistant to the effects of insulin and/or producing less insulin from the pancreas [18,23]. Unlike type 1 diabetes, T2D is strongly associated with lifestyle [25]. Key risk factors for developing T2D are being overweight, having a poor diet, low physical activity levels, smoking, high blood pressure, and high cholesterol [2,22]. The high prevalence of these risk factors in Indigenous peoples can be attributed largely to social determinants of health, presenting a greater challenge for engagement in health-promoting behaviours such as physical activity and maintaining a healthy diet [2,4,7,13]. 

Regular physical activity is a key component of T2D management [26,27,28]. Exercise is a form of structured, deliberate physical activity to maintain or improve physical fitness [29]. Physical activity and exercise can have multiple beneficial effects, including weight loss, increased physical fitness, an increase in sensitivity to insulin, improved mood, improved self-management, improvements in glycaemic control, and a decrease in cardiovascular disease risk [17,30]. There is evidence that community-based, supervised group exercise interventions can improve health outcomes, including glycaemic control, physical fitness, functional capacity, waist circumference, and mental health, for non-Indigenous individuals with T2D [17,30,31]. However, the generalisability of these results to Indigenous populations is limited, as it has been demonstrated that Indigenous input into the design, implementation, and evaluation of health interventions improves Indigenous health outcomes to a greater degree than interventions designed for non-Indigenous populations [32]. Whilst there is some evidence that group-based exercise provides biopsychosocial benefits for healthy Indigenous populations [33], the health impacts of community-based, supervised group exercise interventions for Indigenous populations with T2D have not been sufficiently evaluated. 

This systematic review aims to assess the effectiveness of community-based exercise interventions on health outcomes for Indigenous peoples with T2D. Secondly, we aim to explore the factors which impact participation in community-based exercise among Indigenous peoples.

## 2. Materials and Methods

### 2.1. Study Design 

This systematic review was conducted according to the Preferred Reporting Items for Systematic Reviews and Meta-Analyses (PRISMA) statement [34] and was registered with the International Prospective Register of Systematic Reviews (PROSPERO: ID no. CRD42023414591). 

### 2.2. Search Strategy

The search strategy was developed in consultation with two senior health researchers and a health research librarian. The search was conducted in April 2023 in two stages. The following databases were initially searched: CINAHL, Embase, Informit Indigenous Collection, Medline, and Scopus, but due to limited search results in the domain of interest, PubMed, SportDiscus, and Web of Science databases were added to the search. 

Participants/populationIndigenous adults 18 years of age or older participate in community-based exercise programmes intended to prevent or manage type 2 diabetes.

Intervention(s), exposure(s)Community-based group exercise programmes are intended to prevent or manage type 2 diabetes.

Comparator(s)/controlParticipants who receive usual care. Those who do not receive community-based group exercise programmes intended to prevent or manage type 2 diabetes.

Main outcome(s)Pre and post-health outcomes using behavioural, clinical, or psychosocial measures. Primary outcomes of interest include glycaemic control (measured by HbA1c), body mass index (BMI), waist circumference, physical fitness measures, functional capacity, and psychosocial outcomes, e.g., quality of life (QOL).

Additional outcome(s)Clinician-type, clinician experience, setting of intervention, mode of delivery, and level of adherence to the intervention.

### 2.3. Inclusion-Exclusion Criteria

This review included full-text, published, peer-reviewed literature with original outcome data that reported on the effectiveness of community-based exercise interventions for the management of T2D among Indigenous peoples. The inclusion criteria included lifestyle interventions where exercise was a purposeful, structured, and required part of the program; Indigenous peoples from high-income colonised countries, including Aboriginal and Torres Strait Islander peoples in Australia; Māori and Pacific peoples in New Zealand and the Pacific Islands; American Indian and Alaskan Native peoples in the USA; and First Nations, Inuit, and Métis peoples in Canada. 

Studies were excluded if the intervention was not primarily based in the community. Studies predominantly examining adolescents or women with gestational diabetes were excluded if data for adults with T2D was not disaggregated. Articles published prior to 2002 that were not published in English were also excluded. Studies were also excluded if they were not peer-reviewed and did not include original pre-post intervention outcome data, case series or letters to the editor.

### 2.4. Study Selection

Citations of all relevant literature from the selected databases were exported into Endnote citation software, version 20 [35], and any duplicates were removed. A screen of the titles and abstracts eliminated all studies that did not align with the inclusion criteria. The remaining articles underwent full-text screening by two reviewers (LH and LW) to determine eligibility, with any disagreements settled by a third reviewer (SB). The reference lists of all selected studies were also examined for any additional relevant articles.

### 2.5. Outcome Measures

Outcome measures of interest for this review included pre and post-health outcomes using biomedical or psychosocial measures. Primary outcomes of interest included glycaemic control (measured by HbA1c), body mass index (BMI), waist circumference (WC), physical fitness measures, functional capacity, and psychosocial outcomes, e.g., quality of life (QoL). Additional outcomes of interest were the level of adherence to the intervention and the degree of input Indigenous peoples had into the design and implementation of the intervention.

### 2.6. Quality Appraisal

The quality of selected studies was assessed using the Mixed Methods Appraisal Tool (MMAT) and the Community Engagement Tool (CET) [36,37,38]. 

#### 2.6.1. Mixed Methods Appraisal Tool (MMAT)

The included studies were assessed by two reviewers (LH and SB) using the MMAT [36]. A score of 1 was given for ‘yes’, and 0 for ‘no’, for each of the five assessment categories, producing a total score out of 5. This score was then translated to a percentage and used to determine an overall rating of ‘low’ (0–30%), ‘medium’ (30–70%), or ‘high’ (70–100%) for methodological quality. Any disagreements were settled with discussion and a third reviewer (LW) where necessary.

#### 2.6.2. Community Engagement Tool (CET)

The included studies were scored for the degree of Indigenous community engagement in the study design and implementation by three reviewers (LH, SB, and CC) using the CET. The CET is a modified evaluation tool [37] based on the Australian National Health and Medical Research Council (NHMRC) guidelines for ethical research with Indigenous communities. This tool was implemented in collaboration with an Indigenous research fellow and co-author (CC). The perspective of the Indigenous reviewer was privileged over the other two reviewers for the assessment and overall rating of each study.

The CET includes five criteria: issues identified by the community, Indigenous governance, capacity building, cultural consideration in design, and respecting community experience. The included studies were assessed against these five criteria by three reviewers, and a score out of 5 was determined. Each article was then categorised as weak (0–1), moderate (2–3), or strong (4–5). The full process of using the CET is described elsewhere [37]. 

### 2.7. Data Extraction

All relevant information was extracted into a data extraction table created using Microsoft Excel. Data were exported by one reviewer (LH) and cross-checked by a second (SB). Data of interest included population details, study design, intervention description, and pre and post-measures of relevant health outcomes (as per outcome measures).

### 2.8. Data Synthesis

A thematic analysis of the qualitative data was conducted. Findings involving outcome measures relevant to the research question were categorised into ‘biomedical’ and ‘psychosocial’. Cross-study comparisons were made where possible. The research team found insufficient homogeneity in the studies for a meta-analysis.

## 3. Results

### 3.1. Study Selection

The results of the study selection process, including identification, screening, and inclusion, are reported in Figure 1. A total of 291 articles were identified from the selected databases, and 211 of these were screened using the inclusion and exclusion criteria following the removal of duplicates. Following a title and abstract screen, 21 articles were selected for full-text review, of which three were eligible for inclusion in the review, noting that two of the three publications were about the same study. Reasons for exclusion following full-text screening included: results for Indigenous participants with T2D were not disaggregated; exercise was not a structured and required part of the intervention; the study did not examine health-related outcome measures; no results had been published when the search was conducted. Finally, a reference list screen of the included studies yielded no relevant articles.

### 3.2. Quality Appraisal

The results of the quality appraisal using the MMAT and CET are presented in Table 1. The Shah et al. [12] study was determined to be of high methodological quality, while the two remaining articles were graded to be of medium quality [10,11]. The lower score for the two articles by Sukala et al. [10,11] was predominately due to the low retention rate and compliance level of the intervention. 

Regarding community engagement, two of the included studies were rated moderate using CET [10,11], with one appraised as strong [12]. The article by Shah et al. [12] was given ‘yes’ for criteria 1; issue identified by community; and criteria 3; capacity building, whilst all included studies were scored ‘yes’ for criteria 4; cultural consideration in design; and criteria 5; respecting community experience. Criteria 2, Indigenous governance, was rated ‘no’ across all included studies.

### 3.3. Study Characteristics

The characteristics of the three included studies are summarised in Table 2 and Table 3. Two studies analysed the same intervention with different outcome measures and, as such, were grouped together for the presentation of results [10,11]. These two studies were randomised parallel arm designs conducted with participants of Polynesian descent in New Zealand [10,11], and the remaining study was a prospective cohort design with Native American Zuni Indian participants in New Mexico, USA [12]. Participants in all three studies were predominantly female, with 72% [10,11] and 68% [12]. The median age was consistent across all studies at approximately 49 years of age [10,11,12]. Comorbidities included visceral obesity (BMI = 43.8 ± 9.5 kg/m^2^), reported in two studies [10,11]. The participant retention rate was low for two articles [10,11], with a total of 69% of recruited participants completing the intervention in comparison to the 100% completion rate reported in the third study [12]. 

The exercise interventions implemented across the three studies followed the guidelines of either the American College of Sports Medicine [10,11] or the American Diabetes Association [12]. In adherence to these guidelines, one study required 150 min of exercise a week, including three walking sessions (individual or group) and optional additional aerobic exercise, as part of a larger education-based lifestyle intervention lasting 6 months [12]. The remaining two studies evaluated the effect of resistance versus aerobic exercise conducted in 40–60-min sessions three times weekly across a 4-month period [10,11]. Exercise intensity was reported in two articles [10,11] and included 2–3 sets of 6–8 repetitions until neural fatigue for the resistance exercise group and 65–85% of heart rate reserve (HRR) for the aerobic exercise group.

Cultural consultation was present across all included studies, with all receiving appropriate ethical and cultural approval for study procedures [10,11,12]. The choice of study design was influenced by cultural consultation, as participating communities believed a control group to be unethical [10,11,12], prompting a change from a randomised controlled trial to a randomised parallel arm study in the studies by Sukala et al. [10,11]. Community engagement by Shah et al. [12] also included the intervention being designed and implemented according to information gathered from community focus group consultation, resulting in supportive elements including training and employment of community health representatives (CHRs), provision of transport, home-based testing, class reminders, and individualised exercise and nutrition plans [12].

### 3.4. Outcome of Interventions

Biomedical and psychosocial outcomes measured across studies included HbA1c (recorded as A1c by Shah et al. [12]), glucose, total cholesterol, triglycerides, BMI, body weight, WC, body fat percentage, Patient Activation Measure (PAM), and QoL measured by the Medical Outcomes Trust Short Form-36 Health Survey (SF-36) [10,11,12].

#### 3.4.1. Biomedical Outcomes

Biomedical outcome measures reported across more than one included study are summarised in Table 4. Statistically significant improvement in HbA1c was only reported in one of three studies [12], recording a change from 8.12 ± 2.16 at baseline to 7.39 ± 1.6 after the 6-month intervention (*p* = 0.001). Total cholesterol, glucose, and BMI were significantly reduced in the Shah et al. [12] study, whilst no significant change was found by Sukala et al. [10]. Post-intervention triglycerides were reduced in one study [12] and increased in another secondary to aerobic exercise [10]. This aerobic exercise group also showed a reduction in systolic and diastolic blood pressure after being elevated at baseline. No other relevant biomedical outcomes reached statistical significance. However, when adherence to the resistance or aerobic exercise interventions reached 75% or greater, WC reduced (*p* < 0.001) and body weight tended to improve (*p* = 0.11) [10]. 

#### 3.4.2. Psychosocial Outcomes

Sukala et al. [11] examined the impact of community-based aerobic and resistance exercise interventions on QoL using the SF-36, whilst Shah et al. [12] evaluated the effects of a lifestyle intervention involving education, diet change, and individualised group exercise on PAM levels. Of the eight SF-36 QoL domains, the resistance exercise group showed significant improvements in six domains, whilst the aerobic training group improved in four domains. The Physical Component Summary Score improved in both groups, and the Mental Component Summary Score trended towards improvement (*p* = 0.11 resistance, *p* = 0.14 aerobic) [11]. Pooled analyses of both exercise types identified a moderate to large significant time effect for all QoL domains and summary scales, with the exception of the mental health domain (*p* = 0.09) [11]. Social interaction was acknowledged as an important contributor to these QoL improvements, in particular the 18% increase reported in the social functioning domain [11].

Similarly, post-interventional changes in PAM levels also reached statistical significance, with 58% of participants increasing their level by one or more, 40% maintaining their level, and 2% showing a decrease (*p* < 0.0001) [12]. Despite being unable to demonstrate a correlation between increased patient activation and metabolic improvements, these changes occurred concurrently. The study by Shah et al. [12] concluded that the use of CHRs, point-of-care (POC) testing, and programme individualisation were contributors to participant empowerment by removing barriers to care such as access to care, travel, language barriers, wait times, and a lack of patient–clinician relationships.

## 4. Discussion

The aim of this systematic review was to determine if community-based exercise interventions can improve biopsychosocial health outcomes in Indigenous peoples with T2D. Despite the considerable body of epidemiological research relating to T2D in Indigenous communities [39,40,41,42,43,44,45,46,47] and a comprehensive search strategy, this review was only able to identify three translational studies that were eligible for inclusion [10,11,12]. This dearth of commensurate translational research is indicative of a discrepancy between the proficiency of researchers in identifying concerns and the inadequacy of efforts directed towards implementing culturally appropriate translational solutions [48,49,50]. As such, this review highlights the limited translational research evaluating the health impacts of community-based, supervised group exercise interventions for Indigenous populations with T2D. Furthermore, in support of current recommendations, this review determined that to best facilitate the development of robust, feasible, and culturally safe evidence in this domain, interventions should be adequately contextualised to culture through consistently strong Indigenous governance and community engagement [37,38,50,51].

With consideration of the small number of included studies, this review found that community-based exercise can improve glycaemic control [12], anthropometric measures [10,12], blood lipids [12], haemodynamic measures [10], patient activation [12], and quality of life [10] in Indigenous peoples with T2D [17,30,31]. However, upon comparing the two featured interventions [10,11,12], these adaptations appear to be largely dependent on successful cultural contextualisation of the intervention and subsequently, greater participation in the planned intervention [37,38,50,51,52,53,54]. Shah et al. [12] demonstrated that biomedical outcomes, including glycaemic control, can be improved with group walking and aerobic exercise programmes as part of a holistic lifestyle intervention, including education and individualised nutrition advice. This community and home-based, CHR-implemented intervention was created and implemented to remove community-specific barriers to diabetes care and improve patient activation following community focus group consultation [12,55]. As such, with the exception of Indigenous governance (due to inadequate levels of Indigenous leadership), this intervention met all other CET criteria and was determined to have strong community engagement [12,37,38]. In conjunction with wider literature [37,38,50,51,52,53], it can, therefore, be hypothesised that the strong community engagement contributed to the 100% compliance rate and the significant improvement in biomedical outcomes [12]. 

In comparison, in the study by Sukala et al. [10], several post-intervention biomedical outcome measures were inconsistent with expectations despite the length [28,56], frequency, duration [27,28,57], and mode of exercise [58,59] being appropriate to elicit significant change [27,28,57]. The lack of improvement in most biomedical outcomes, including HbA1c, was likely due to an interaction of several factors, including participant characteristics, insufficient contextualisation, and low participation in the intervention [10,54,60]. Participants’ obesity levels and high baseline HbA1c may have required a longer intervention with a higher frequency and duration of exercise sessions to obtain these adaptations [54,61]. Furthermore, with less than 70% participation in the exercise intervention, for many participants, the exercise frequency would have fallen below physical activity guideline recommendations [10,26,27,57]. When adherence reached 75%, clinically meaningful adaptations in anthropometric measures were demonstrated [10]. The low participation can be attributed, at least in part, to the lower community engagement and subsequent insufficiency of the intervention to meet participants’ cultural and healthcare needs when compared to the Shah et al. intervention [12]. 

Although biomedical improvements were limited, post-intervention quality scores by Sukala et al. [10] approached or surpassed levels reported for the general population (of New Zealand), aligning with strong evidence that exercise can improve health-related quality of life in healthy adult populations [62] and non-Indigenous populations with T2D [17,63]. This is of particular significance given the low baseline SF-36 scores [10] and low quality of life associated with both T2D diagnosis and Indigenous Polynesian peoples [10,64]. Sukala et al. [10] also acknowledged the importance of the socialisation aspect of community-based group exercise, as evidenced by the 18% improvement in the SF-36 Social Functioning domain. Social interaction in a familiar community environment has been shown to improve quality of life and help facilitate exercise participation in both Polynesian and non-Indigenous populations [10,53,65]. 

Participation in exercise and other healthy behaviours may be related to the level of patient activation [12,66,67,68]. The strong community engagement in the study facilitated the increase in patient activation through the removal of the barriers that were identified to inhibit patient empowerment [12]. In particular, the use of CHRs was a vital contributor to participant empowerment by removing language barriers and the lack of patient–clinician relationships [12,53].

This review was limited due to the small number of eligible studies that fit the inclusion criteria and the variation of outcome measures used across the included studies. As such, outcome comparison across studies was limited, and there was insufficient data to perform a meta-analysis. This review included two studies reported in three papers. One study was reported in two papers. Secondly, selection bias may have occurred as a result of expanding the search strategy and inclusion criteria. Further research on community-based exercise interventions in Indigenous peoples with T2D using biomedical and psychosocial health outcomes would, therefore, allow for more in-depth analysis and refinement of the characteristics of effective interventions.

## 5. Conclusions

This systematic review highlights the paucity of literature evaluating community-based group exercise interventions for Indigenous adults with T2D. Despite the limited available research, this review determined that community-based exercise interventions may improve health outcomes for Indigenous adults with T2D when conducted in a culturally safe manner. Furthermore, strong Indigenous community engagement may increase intervention adherence and, therefore, effectiveness by removing barriers to diabetes care identified by the community. Additional research with Indigenous involvement and governance is required to determine the extent of this effect and work towards narrowing the gap in Indigenous and non-Indigenous health outcomes in this domain.

## Figures and Tables

**Figure 1 ijerph-21-00290-f001:**
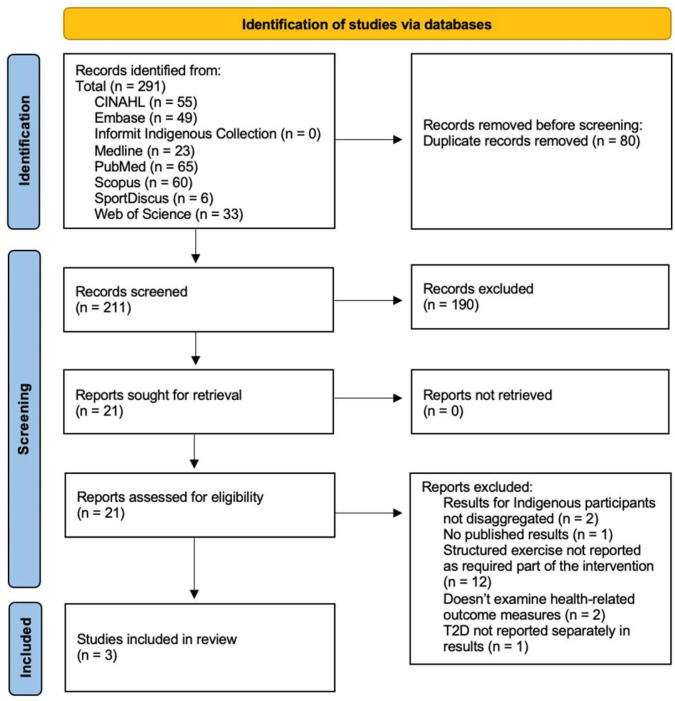
Preferred Reporting Items for Systematic Reviews (PRISMA) flow diagram [34].

**Table 1 ijerph-21-00290-t001:** Final scores and quality rating of MMAT and CET quality appraisal with CET criteria scoring.

Criteria	Sukala et al. 2012 [10]	Sukala et al. 2013 [11]	Shah et al. 2015 [12]
**MMAT**			
Total (%)	40%	40%	80%
Rating	Medium	Medium	High
**CET**			
Issue Identified by the Community	0	0	1
Indigenous Governance	**0**	**0**	**0**
Capacity Building	**0**	**0**	**1**
Cultural Consideration in Design	**1**	**1**	**1**
Respecting Community Experience	**1**	**1**	**1**
Total/5	2	2	4
Rating	Moderate	Moderate	Strong

MMAT, Mixed Methods Appraisal Tool; CET, Community Engagement Tool.

**Table 2 ijerph-21-00290-t002:** Summary characteristics of included studies and participants.

Reference (Author, Year)	Aim	Design	Country	Indigenous Population	Program	Participants				
						Included (n)	Completed (n)	Sex (%)	Age (Years)	Co-Morbidities
(i) Sukala et al., 2012 [10] (ii) Sukala et al., 2013 [11]	(i) To evaluate the effectiveness of two exercise modalities for improving glycosylated hemoglobin (HbA1c) and associated clinical outcomes in Polynesian adults diagnosed with type 2 diabetes and visceral obesity (ii) To evaluate the differential effect of 2, group-based exercise modalities on quality of life (QoL) in indigenous Polynesian peoples with type 2 diabetes (T2DM) and visceral obesity	Randomised parallel arm	New Zealand	Polynesian descent (self-identified): New Zealand Māori (10), Cook Islands Māori (3), Samoan (2), Fijian (1), Tokelauan (1), Tongan (1)	Resistance vs. Aerobic Exercise Intervention	n = 26	n = 18	F: 72% M: 28%	49 ± 5	Visceral Obesity
Shah et al., 2015 [12]	To evaluate the use of a novel chronic disease prevention approach for improving clinical outcomes and health-related quality of life in Zuni Indians with chronic disease	Prospective cohort	Zuni Pueblo, New Mexico, USA	Native American ancestry: Zuni Indians	Lifestyle Intervention: group educational classes and home-based teaching, point of care testing, and individualized exercise and nutritional programs	n = 60	n = 60	F: 68% M: 32%	49.4 ± 12.0	None Reported

**Table 3 ijerph-21-00290-t003:** Summary of exercise intervention characteristics, outcome measures, and results of included studies.

Reference (Author, Year)	Exercise Type	Exercise Characteristics	Exercise Intensity	Exercise Volume	Length of Intervention	Intervention Compliance	Outcome	
							Biomedical	Psychosocial
(i) Sukala et al., 2012 [10] (ii) Sukala et al., 2013 [11]	Resistance Exercise	Resistance: Eight exercises using machine weights targeting all the major muscle groups of the body: seated leg press, knee extension, knee flexion, chest press, lat pulldown, overhead press, biceps curl, and triceps extension	2–3 sets of 6–8 repetitions with 1 min rest between sets and exercises. Load increased by 5% when 10 repetitions could be performed	40–60 min per session 3 sessions per week	16 weeks	67 ± 18%	(i) HbA1c (%), HOMA2-IR index, McAuley index, Glucose (mmol/L), Insulin (pmol/L), C-peptide, Free fatty acids (mEq/L), Log C-reactive protein, Adiponectin, Blood Lipids (mmol/L): Total cholesterol, HDL cholesterol, LDL cholesterol, Triglycerides, Body weight (kg), BMI (kg/m^2^), Waist circumference (cm), Body fat (%), SBP & DBP	(ii) Quality of life: Medical Outcomes Trust Short Form-36 Health Survey (v. 1.0) (SF-36)
	Aerobic Exercise	Aerobic: Exercises on a cycle ergometer	First 2 weeks: progression from 65% to 85% of participants heart rate reserve (HRR) Remaining program: 85% of HRR			73 ± 19%		
Shah et al., 2015 [12]	Walking Program Aerobic Exercise	Exercise in addition to normal activity within peer groups of 4–6 individuals: walking at least 3× weekly (on their own or within CHR organised walking groups), and optional additional aerobic exercise at Zuni Health Facility	Not Reported	150 min weekly 3× weekly (walking)	6 months	100%	A1c%, Glucose(mg/dL), Total Cholestrol (mg/dL), Triglyceride (mg/dL) Blood Urea Nitrogen/BUN (mg/dL), Creatinine (mg/dL), Urine Albumin-to-Creatinine ratio/UACR (mg ALB/g CR), Uric Acid (mg/dL), BMI	PAM levels: Patient Activation Measure (PAM)-short form PAM questionnaire

Abbreviations: CHR, Community Health Representative; HbA1C/A1c, haemoglobin A1C; HOMA2-IR, Homeostatic Model Assessment for Insulin Resistance; BMI, Body Mass Index; SBP, Systolic Blood Pressure; DBP, Diastolic Blood Pressure; QoL, Quality of Life; PCS, Physical Component Summary.

**Table 4 ijerph-21-00290-t004:** Summary of biomedical outcomes.

Author	Year	Group	HbA1c (%)				Glucose (mmol/L)				Total Cholesterol (mmol/L)				BMI (kg/m^2^)			
			Pre	Post	% Change	*p* Value	Pre	Post	% Change	*p* Value	Pre	Post	% Change	*p* Value	Pre	Post	% Change	*p* Value
Sukala et al.	2012 [10]	RET	10.7 ± 2.1	10.6 ± 2.4	−0.1 ± 1.1	0.86	9.5 ± 3.5	11.4 ± 4	1.9 ± 3.2	0.17	4.9 ± 1.5	4.5 ± 1	−0.4 ± 0.9	0.21	42.7 ± 12.1	42.7 ± 11.7	0.0 ± 1.1	0.91
		AET	8.9 ± 1.9	8.8 ± 2.1	−0.1 ± 0.6	0.6	10.2 ± 3.3	10.4 ± 2.9	0.2 ± 1.6	0.72	4.5 ± 0.4	4.7 ± 0.4	0.3 ± 0.6	0.22	45 ± 6.5	44.5 ± 6.9	−0.5 ± 1.3	0.32
Shah et al.	2015 [12]		8.12 ± 2.16	7.39 ± 1.6	Not Reported	0.001	8.8 ± 3.9	7.5 ± 2.5	Not Reported	0.0003	4.1 ± 1.0	3.8 ± 0.8	Not Reported	0.003	33.8 ± 8.4	32.4 ± 8.2	Not Reported	0.001

RET, Resistance Exercise Training; AET, Aerobic Exercise Training; BMI, Body Mass Index.

## Data Availability

Data can be made available upon request.

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
