# Peer review of "The Effect of Community-Based Exercise on Health Outcomes for Indigenous Peoples with Type 2 Diabetes: A Systematic Review"

_ijerph, 2024, doi:10.3390/ijerph21030290_

Round 1

Reviewer 1 Report

Comments and Suggestions for Authors

Dear authors,

congratulations for your work about "The Effect of Community-Based Exercise on Health Outcomes for Indigenous Peoples with Type 2 Diabetes". Your goal was to " determine if health outcomes in Indigenous peoples with T2D could be improved by community-based exercise programs and the features of those programs that best meet their needs.". 

In general your manuscript is well written but there are procedures that must be performed. 

1. Your topic is very specific and that must be taken with causion.

2. You should perform PICOS criteria.

3. You should perform risk of bias and certain of evidence.

4. In discussion you state "that community-based exercise can improve glycaemic control [12], anthropometric measures [10,12], blood lipids [12], haemodynamic measures [10], patient activation [12], and quality of life [10] in Indigneous peoples with T2D [17,30,31]." but you do not compare this specific results with other studies without Indigenous subjects. It would be interesting to add sentences comparing booth results.

congratulations

Author Response

Thank you for taking the time to review our paper.

Please find our responses to your comments with references to the tracked changes where needed.

Section

Reviewer comment

Response to reviewer comment

General

1.       Your topic is very specific and that must be taken with causion.

2. You should perform PICOS criteria.

3. You should perform risk of bias and certain of evidence.

1.       Thank you

2.       Thanks- we agree and have done a PICO criteria as a part of our PROSPERO process.

3.       We agree and have used the MMAT and CET tools in this study

Discussion

 you state "that community-based exercise can improve glycaemic control [12], anthropometric measures [10,12], blood lipids [12], haemodynamic measures [10], patient activation [12], and quality of life [10] in Indigneous peoples with T2D [17,30,31]." but you do not compare this specific results with other studies without Indigenous subjects. It would be interesting to add sentences comparing booth results.

The research team found insufficient homogeneity in the studies for a meta-analysis. We have included this detail at the end of the Data Synthesis section

Reviewer 2 Report

Comments and Suggestions for Authors

Dear Author(s),

Thank you for your systematic review manuscript, which aim was to determine if community-based exercise interventions can improve bio-psycho-social health outcomes in Indigenous peoples with T2D.

The topic is important and clinically relevant. However, I have few concerns that need to be disclosed and without those explanation in detail exploration of the quality of your work can not be made.

Firstly, you need to include keywords and logical operators per individual databases used for identification. It would be nice if you can include the number of citations identified using the latter through each database separately (so the reviewer and reader can re-run the searches).

Secondly, you should describe in more detail the inclusion and exclusion criteria of the studies. Did you use only full-text or abstracts also? What types of studies have you excluded (case reports, case seria, letter to the editor, etc.).

Thirdly, it would be nice if you can mention the ratio of similarity between screeners in the screening process and the percentage of citations (or number) for which the citation was need to be weighed by the third reviewer.

In addition, you have two very similar intervention studies from the same author (2012 and 2013 yr.), do you consider those as salami publications? If so, it should be mentioned within discussion and limitation since duplicate in consclusions and bias may arise that way.

Lastly, please include the references of novel date within your reference list to demonstrate that this article is relevant even today and up to date (at least 20% references should be published in the last 5 yr.).

Best regards, Peer reviewer

Author Response

Thank you for taking the time to review our paper.

Please find our responses to your comments with references to the tracked changes where needed.

Reviewer comment

Response to reviewer comment

Firstly, you need to include keywords and logical operators per individual databases used for identification. It would be nice if you can include the number of citations identified using the latter through each database separately (so the reviewer and reader can re-run the searches).

We have included the search string in the manuscript in the Search Strategy section. Apologies for the omission!

Secondly, you should describe in more detail the inclusion and exclusion criteria of the studies. Did you use only full-text or abstracts also? What types of studies have you excluded (case reports, case seria, letter to the editor, etc.).

Thank you for this suggestion. Please find more detail added to the Inclusion and exclusion section.

Thirdly, it would be nice if you can mention the ratio of similarity between screeners in the screening process and the percentage of citations (or number) for which the citation was need to be weighed by the third reviewer.

Any disagreement regarding inclusion or exclusion were resolved through discussion until consensus was reached. The Indigenous researcher’s perspective was privileged throughout.

In addition, you have two very similar intervention studies from the same author (2012 and 2013 yr.), do you consider those as salami publications? If so, it should be mentioned within discussion and limitation since duplicate in consclusions and bias may arise that way.

Thank you. We do not believe it is salami publishing but have noted in the Results section of the manuscript that it is two separate papers about the same study.

Lastly, please include the references of novel date within your reference list to demonstrate that this article is relevant even today and up to date (at least 20% references should be published in the last 5 yr.).

We respectfully disagree. The scope of this review was 2002-2022.

Round 2

Reviewer 1 Report

Comments and Suggestions for Authors

Dear authors,

congratulations for your work.

Author Response

Thanks for your response to our revisions. We appreciate the feedback. 

Reviewer 2 Report

Comments and Suggestions for Authors

Dear Author(s),

Thank you for your modification. The quality is now improved.

I still believe those 2 articles included are salami publications and this should be somehow commented within discussion or as a limitation because duplication bias may arise.

Regarding the publications of novel date - there is definitely need to include them for introduction and discussion section.

Minor revision is still definitely needed.

Best regards, Peer reviewer

Author Response

Thank you for taking the time to review our paper.

Please find our responses to your comments with references to the tracked changes where needed.

Reviewer comment

Response to reviewer comment

Thank you for your modification. The quality is now improved.

Thank you.

I still believe those 2 articles included are salami publications and this should be somehow commented within discussion or as a limitation because duplication bias may arise.

Thank you- text added to the limitations section.

Regarding the publications of novel date - there is definitely need to include them for introduction and discussion section.

The comment is unclear. Please clarify the request.

Minor revision is still definitely needed.

Thank you